

# Fast and automated identification of reactions with low barriers using meta-MD simulations

Maria H. Rasmussen and Jan H. Jensen

Department of Chemistry, University of Copenhagen, Copenhagen, Denmark

## ABSTRACT

We test our meta-molecular dynamics (MD) based approach for finding low-barrier (<30 kcal/mol) reactions on uni- and bimolecular reactions extracted from the barrier dataset developed by *Grambow, Pattanaik & Green (2020)*. For unimolecular reactions the meta-MD simulations identify 25 of the 26 products found by *Grambow, Pattanaik & Green (2020)*, while the subsequent semiempirical screening eliminates an additional four reactions due to an overestimation of the reaction energies or estimated barrier heights relative to DFT. In addition, our approach identifies 36 reactions not found by *Grambow, Pattanaik & Green (2020)*, 10 of which are <30 kcal/mol. For bimolecular reactions the meta-MD simulations identify 19 of the 20 reactions found by *Grambow, Pattanaik & Green (2020)*, while the subsequent semiempirical screening eliminates an additional reaction. In addition, we find 34 new low-barrier reactions. For bimolecular reactions we found that it is necessary to "encourage" the reactants to go to previously undiscovered products, by including products found by other MD simulations when computing the biasing potential as well as decreasing the size of the molecular cavity in which the MD occurs, until a reaction is observed. We also show that our methodology can find the correct products for two reactions that are more representative of those encountered in synthetic organic chemistry. The meta-MD hyperparameters used in this study thus appear to be generally applicable to finding low-barrier reactions.

## INTRODUCTION

Understanding how molecular systems react, *i.e.* which reactions are possible under what conditions, is an essential part of chemical research and computational methods for exploring reaction space in an automated manner are continually being proposed (*Zimmerman, 2013*; *Suleimanov & Green, 2015*; *Habershon, 2016*; *Varela, Vázquez & Martínez-Núñez., 2017*; *Kim et al., 2018*; *Dewyer, Argüelles & Zimmerman, 2018*; *Robertson & Habershon, 2019*; *Unsleber & Reiher, 2020*; *Lavigne et al., 2020*; *Shannon et al., 2021*; *Koerstz, Rasmussen & Jensen, 2021*; *Shannon et al., 2021*; *Van de Vijver & Zádor, 2020*; *Young et al., 2021*; *Zimmerman, 2015*; *Maeda, Taketsugu & Morokuma, 2014*; *Grimme, 2019*; *Wang et al., 2014*). These methods can be divided into three different categories:

Corresponding author
Jan H. Jensen, jhjensen@chem.ku.dk

(semi-) exhaustive searches (*Kim et al., 2018*; *Habershon, 2016*; *Zimmerman, 2013*; *Suleimanov & Green, 2015*; *Koerstz, Rasmussen & Jensen, 2021*), reaction template methods (*Shannon et al., 2021*; *Van de Vijver & Zádor, 2020*; *Young et al., 2021*; *Zimmerman, 2015*; *Maeda, Taketsugu & Morokuma, 2014*), and meta-molecular dynamics (meta-MD) based approaches (*Shannon et al., 2021*; *Grimme, 2019*; *Koerstz, Rasmussen & Jensen, 2021*; *Wang et al., 2014*). Examples of (semi-) exhaustive searches include graph enumeration of products (*Kim et al., 2018*; *Habershon, 2016*; *Zimmerman, 2013*; *Suleimanov & Green, 2015*; *Koerstz, Rasmussen & Jensen, 2021*) and enumeration of reaction coordinates (*Zimmerman, 2015*; *Maeda, Taketsugu & Morokuma, 2014*). For these methods the size of the search space, and hence the computational cost, grows quickly with the size of the molecules. The reaction template approaches lie at the other extreme in terms of computational efficiency, and work by investigating only pre-determined reaction types. Though efficient and used extensively in atmospheric and combustion chemistry, this approach can be hard to generalise to other areas such as synthetic organic chemistry, though a recent attempt is encouraging (*Young et al., 2021*). The meta-MD approaches explore reactivity *via* biasing potentials that force reactions and exploration of conformational space. The meta-MD approach can also be combined with (semi-) exhaustive search methods as shown by *Lavigne et al. (2020)*. In the non-exhaustive approaches the key question is whether they identify *all* relevant reactions for the problem at hand. At room temperature, this typically means all reactions with barriers less than ca 30 kcal/mol given the typical accuracy of quantum chemical calculations.

Recently, we demonstrated that a combination of product generation using meta-MD and barrier estimation + transition state (TS) guess generation using the RMSD-PP method (*Grimme, 2019*), both relying on semiempirical GFN2-xTB (*Bannwarth, Ehlert & Grimme, 2019*) calculations, could efficiently suggest the three lowest barrier elementary reactions of 3-hydroperoxy-propanal (*Koerstz, Rasmussen & Jensen, 2021*). The results depend strongly on the hyperparameters that control the biasing potential in the meta-MD and it is unclear how well this hyperparameter set generalizes to other unimolecular reactants and to bimolecular reactions in general. Here we test the performance of this hyperparameter set on elementary unimolecular reactions involving 163 reactant molecules and 20 elementary bimolecular reactions, both extracted from the recently published database of elementary reactions involving H, C, N, and O on two different DFT levels (*Grambow, Pattanaik & Green, 2020*). In addition we test the method on two multi-step reactions related to organic synthesis.

## METHODS

Our method for predicting the kinetically important elementary reactions (*Koerstz, Rasmussen & Jensen, 2021*) is based on three steps (Fig. 1): (a) the generation of possible single-step products which is based on meta-molecular dynamics (meta-MD) simulations followed by (b) screening of the proposed products based on reaction energies and estimated barrier heights computed at the semiempirical (GFN2-xTB) level of theory followed by (c) validation at the DFT level of theory. In this study we use a cutoff of

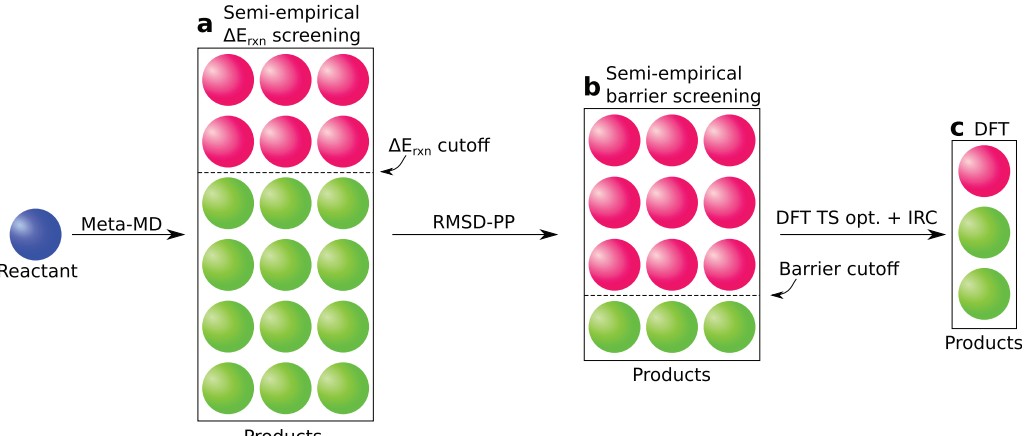

**Figure 1 Schematics of the method used to predict low-barrier reactions employed in this study.** A cutoff value of 40 kcal/mol is used to screen for low-barrier reactions (circles below the dotted line). Abbreviations are meta-MD, meta-molecular dynamics; $\Delta E_{rxn}$, electronic reaction energy; RMSD-PP, root-mean-square deviation-push/pull; DFT, density functional theory; TS, transition state; IRC, intrinsic reaction coordinate.

40 kcal/mol when screening reaction energies and barrier estimates as a compromise between the accuracy of the energies and the number of reactions to be checked.

Our approach to product generation is based on the meta-MD approach by *Grimme (2019)* which is a way of increasing the likelihood of a reaction occurring during an MD simulation by penalising to previously visited structures. This is done by adding additional terms to the energy and gradient that depend on the RMSD from previously visited structures during the current MD simulation or previous MD simulations (selected by the user). As described in *Koerstz, Rasmussen & Jensen (2021)* the additional energy terms depend on the hyper-parameters $k_{push}$, $\alpha$, and $s$ and we use values that were optimised to promote unimolecular chemical reactions (*Koerstz, Rasmussen & Jensen, 2021*) ($k_{push} = 0.05E_h$, $\alpha = 0.3$ Bohr$^{-2}$, $s = 0.8$) along with an additional set found empirically as part of this study ($k_{push} = 0.03E_h$, $\alpha = 0.7$ Bohr$^{-2}$, $s = 0.6$). For the bimolecular reactions, we change the procedure slightly. Initial runs showed greatly increased run times compared to single-fragment reactants using the original hyperparameters. To decrease run times we decrease $s$ (which scales the size of the molecular cavity) by 0.02 every 5 ps as long as no reaction has occurred, thereby forcing the reactant molecules closer together.

We generate different random Cartesian coordinates ("embedding" in RDKit) for the reactants for each of the meta-MD runs based on their SMILES string. If the reactant consists of multiple molecules, they are first embedded randomly on top of each other using RDKit (*Landrum, 2020*) with a subsequent force field optimization of each fragment. The second molecule is then moved in a random direction by a distance, $d$:

$$d = 0.5 \cdot (D_{max,1} + D_{max,2}) + 2\text{Å} \tag{1}$$

where $D_{max, i}$ is the maximum distance between any two atoms in molecule $i$. The coordinates of the reactants are energy minimised with GFN2-xTB before starting the

meta-MD. If a change in atomic connectivity is detected the meta-MD simulation is skipped and the barrier estimation is done using the unoptimised reactant structure. Bond detection is done by xyz2mol based on the overlap density from an extended Hückel calculation, which must be greater than 0.2 for a bond. We perform 100 meta-MD simulations for each reactant unless otherwise noted. Our algorithm checks for changes in atomic connectivity every 5 ps and the simulation is stopped when this is detected. The output of the meta-MD simulations is then a list of all unimolecular reactions and a database of the reactant and product structures.

Barrier estimates and transition state (TS) guess structures are based on the RMSD-PP procedure by *Grimme (2019)* and is described in detail in *Rasmussen & Jensen (2020)* and *Koerstz, Rasmussen & Jensen (2021)*. The RMSD-PP method locates TS guess structures by interpolating between reactants and products *via* biasing potentials and has roughly the same computational cost as a geometry optimisation. Instead of embedding the reactant and product structures from the SMILES saved during the product generation as done in *Koerstz, Rasmussen & Jensen (2021)*, we use the optimized structures from the meta-MD procedure as input to the RMSD-PP procedure. For the barrier estimate, the RMSD-PP is run five times, and the lowest barrier estimate is used, as described in *Koerstz, Rasmussen & Jensen (2021)*, except two of the five runs are done starting the procedure from product → reactant instead of reactant → product. If available, different conformers of the reactant and product are used in each of the five runs. If the reaction is found in five or more of the 100 meta-MD runs, the reactant/product structures are extracted from a different meta-MD simulation in each of the five RMSD-PP calculations. If the reaction is found four times during the meta-MD simulations, two of the RMSD-PP calculations use reactant/product structures from the same meta-MD simulation, while the remaining three RMSD-PP calculations use reactant/product structures from three different runs and so on for cases where the reaction is found three, two or one time during the 100 meta-MD runs. For the computation of reaction energies and barriers we need energies of the reactant and products. The lowest energy structure encountered for each molecule (each unique canonical SMILES) during the geometry optimizations that follow the meta-MD runs is used.

The last step of the procedure is the DFT-refinement (ωB97X-D/def2-TZVP) of the reactions found at the semiempirical level of theory that have reaction energies and barriers below 40 kcal/mol. Again, our validation procedure is as described in *Rasmussen & Jensen (2020)* and *Koerstz, Rasmussen & Jensen (2021)* except that we only test one of the five possible TS guess structures that can be extracted from the five RMSD-PP runs in order to reduce the computational cost of the DFT part of the procedure. The barriers and reaction energies at DFT level of theory are computed as stated in *Grambow, Pattanaik & Green (2020)*, adding zero-point vibrational energies to the electronic energies of reactant, product and transition states (TSs) before calculating the barrier as the energy difference between TS and reactant and reaction energy as the energy difference between product and reactant.

All Density Functional Theory (DFT) calculations are performed using Gaussian 16 (*Frisch et al., 2016*). The meta-MD calculations and the RMSD-PP barrier estimates are

performed with version 6.1.4 of the `xtb` program (*Grimme, 2019*) using the GFN2-xTB method (*Bannwarth, Ehlert & Grimme, 2019*). All structure-to-SMILES and structure-to-adjacency matrix (AC) ($N_{atoms} \times N_{atoms}$ dimensional matrix with elements either 1 or 0 depending on whether the atom-pair is bound or not) conversions are done using `xyz2mol` (*Jensen, 2021*).

## RESULTS AND DISCUSSION

### Unimolecular reactions

#### *The low-barrier reaction dataset*

The low-barrier dataset used in this study is extracted from the dataset created by *Grambow, Pattanaik & Green (2020)*. The reactants in the Grambow dataset consists of all molecules in the GDB-7 dataset (*Ruddigkeit et al., 2012*) with less than seven heavy atoms plus ca. 430 randomly selected molecules with seven heavy atoms. Reactions and the corresponding transition states (TSs) are located by performing several hundred single-ended growing string method (GSM (*Zimmerman, 2015*)) searches from each reactant at the B97-D3/def2-mSVP level of theory, followed by TS refinement at the ωB97X-D3/def2-TZVP level of theory. The result is 16,365 and 11,961 unimolecular reaction barriers at the B97-D3/def2-mSVP and ωB97X-D3/def2-TZVP level of theory, respectively. The corresponding number of low-barrier reactions with barriers below 30 kcal/mol is 199 and 30, involving 163 (Fig. S1) and 27 different reactants, respectively. We would thus expect that applying our meta-MD search to these 163 reactants (Fig. S1) should identify these 30 reactions (Table S1) if we use ωB97X-D3/def2-TZVP for the DFT refinement.

Since the D3 dispersion correction is not available with the ωB97X functional in Gaussian16 we start by reoptimising the 30 TSs at the ωB97X-D/def2-TZVP level of theory and verifying them by performing intrinsic reaction coordinate (IRCs) calculations. For two reactions (R1108 and R7201, using the notation of *Grambow, Pattanaik & Green, 2020*) the IRCs go to the correct reactant, but to a different stereoisomer. Since that barrier is below 30 kcal/mol we use the newly found structure to initiate the meta-MD instead of the one proposed by *Grambow, Pattanaik & Green (2020)*. For five reactions (R1084, R2399, R2523, R6490, and R18816), the IRC does not lead to the reactant proposed by *Grambow, Pattanaik & Green (2020)*. We subsequently find the TS for R2399 using our meta-MD approach, but we exclude R1084, R2523, R6490, and R18816 from our low-barrier dataset. Thus, we expect that applying our meta-MD search to the 163 reactants described above should identify 26 reactions with barriers below 30 kcal/mol at the ωB97X-D/def2-TZVP level of theory.

#### *Meta-MD based search for low-barrier reactions*

Using the default hyperparameter set ($k_{push} = 0.05$, $\alpha = 0.3$ and $s = 0.8$) we find 20 of the 26 reactions. For three of the reactions (R3725, R7207, and R8701) the meta-MD failed to generate the corresponding product structures. Another reaction (R2514) is eliminated due to having a GFN2-xTB reaction energy of 53 kcal/mol, which is significantly higher than the corresponding ωB97X-D3/def2-TZVP-value of -3 kcal/mol. The final two reactions (R4612 and R9011) are eliminated due to high RMSD-PP estimated barrier

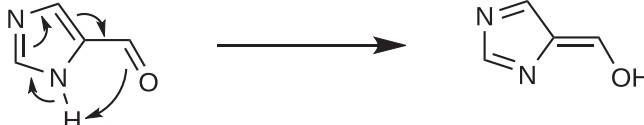

**Figure 2 The reaction not found by either meta-MD hyperparameter sets.** The reaction energy is $\Delta E = 19$ kcal/mol and the barrier is $\Delta E^{\dagger} = 27$ kcal/mol calculated with ωB97X-D3/def2-TZVP (*Grambow, Pattanaik & Green, 2020*).

heights of 48 and 47 kcal/mol, respectively - considerably higher than the corresponding ωB97X-D3/def2-TZVP-values of 26 and 30 kcal/mol. The actual barrier heights at the GFN2-xTB level are 42 and 45 kcal/mol, which indicates that the problem lies primarily with the GFN2-xTB method itself and not the barrier-estimation method. If we perform additional meta-MD simulations with a slightly different hyperparameter set ($k_{push} = 0.03$, $\alpha = 0.7$, $s = 0.6$) we locate R3725 and R7207, but R3725 is subsequently eliminated due to a high reaction energy (42 kcal/mol) compared to a DFT value of −21 kcal/mol. So, using two sets of hyperparameters, the meta-MD simulations identify 25 of the 26 low-barrier reactions reactions found by *Grambow, Pattanaik & Green (2020)*, but four of these are subsequently eliminated due to overestimated reaction energies and estimated barrier heights by GFN2-xTB. Only R8701 (Fig. 2) is not found by the meta-MD, presumably due to a combination of high barrier and reaction energy (27 and 19 kcal/mol at the DFT level) plus a relatively small change in structure on going from reactants to products, resulting in a weak biasing potential.

The four false negatives that result from errors in the GFN2-xTB reaction energies and estimated barrier height (R2514, R3725, R4612, and R9011) all contain a N-N triple bond in the products, indicating a systematic error in the GFN2-xTB method. In the case of reaction energies such errors can be efficiently corrected by, for example, the connectivity-based hierarchy method (*Sengupta & Raghavachari, 2017*; *Kromann et al., 2018*). Another option is simply to use DFT instead of GFN2-xTB for the subsequent screening. The meta-MD approach produces between one and 52 reactions per reactant (Fig. 3) which is practical to check with DFT, and certainly with DFT//GFN2-xTB single point calculations. In our current approach 1,257 and 2,392 reactions are eliminated based on semiempirical reaction energies and estimated barriers, respectively, so that only 316 reactions are checked with DFT out of a total of 3,965 candidate reactions.

Our meta-MD based approach also identifies 10 low-barrier reactions, shown in Fig. 4, not found by *Grambow, Pattanaik & Green (2020)*, as well as 26 new reactions with barriers above 30 kcal/mol (Tables S2 and S3). All low-barrier reactions involve new reactants not represented in the low-barrier dataset, which indicates that the reactions in that dataset are likely the ones with the lowest possible barriers for each reactant. Eleven of the new high-barrier reactions (Table S2) have barriers that are lower than the lowest barrier found by *Grambow, Pattanaik & Green (2020)* for those reactants. Fourteen of the 36 new reactions (N6, N11, N13, N14, N16, N17, N18, N22, N23, N24, N25, N29, N35, and N36) are found by *Grambow, Pattanaik & Green (2020)* at the B97-D3/def2-mSVP level of theory but these TS structures are apparently not of sufficient quality for the

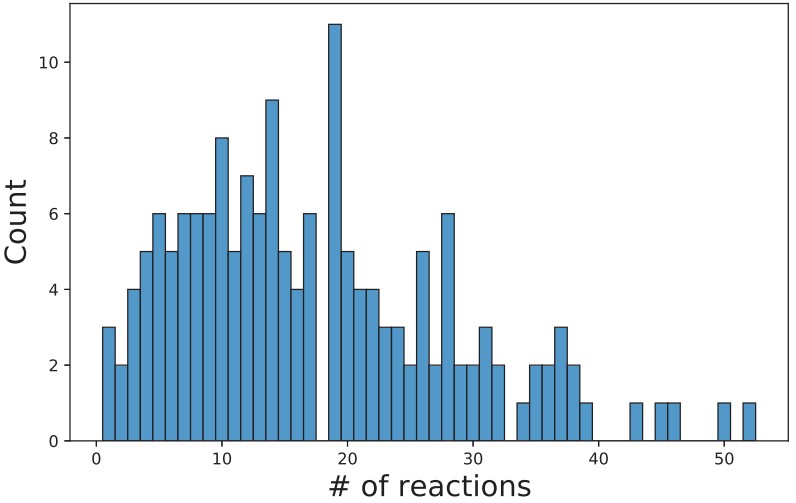

**Figure 3 Distribution of the number of different reactions found during the 100 meta-MD runs (with default hyperparameter set) for the 163 single-fragment reactants shown in Fig. S1.** The average is 17.4 reactions/reactant.

ωB97X-D3/def2-TZVP refinement. For the remaining reactions, it is not clear whether the new reactions are not found by *Grambow, Pattanaik & Green (2020)* due to the growing string method (GSM) itself or the more approximate level of theory used for the GSM calculations. One reaction (N32 in Table S3) corresponds to a change in chirality due to a hydrogen transfer and such reactions do not appear to have been reported by *Grambow, Pattanaik & Green (2020)*, so it is possible that they found it but did not report it.

Two of the new low-barrier reactions (N4 and N7) proceed without a barrier at the GFN2-xTB level of theory, while the DFT barriers are 21 and 12 kcal/mol, respectively and both reactions are endothermic at the DFT level of theory. Both reactions involve a N–N triple bond, though that is also the case for several other new low barrier reactions that were identified successfully.

## Bimolecular reactions

*Grambow, Pattanaik & Green (2020)* only searched for elementary reactions starting from single reactant molecules, but they found a lot of products with two fragments. From these back-reactions, we extract a set of 20 target reactions with barriers below 30 kcal/mol, where two molecules react to create a single molecule (Table S4). The barriers range between 8 and 28 kcal/mol and reaction energies range between −58 and 8 kcal/mol. Unlike for the unimolecular reactants above, these bimolecular reactants were not subjected to a thorough reaction discovery search so it is more likely that there will be additional reactions with barriers below 30 kcal/mol. As with the unimolecular reactions above, we re-optimize the TS structures provided by *Grambow, Pattanaik & Green (2020)* at the ωB97X-D/def2-TZVP level of theory and confirm that they connect the stated reactant and product with an IRC.

Again, we first run 100 meta-MD simulations for each of the 20 reactant pairs with the primary hyperparameter set ($k_{push} = 0.05$, $\alpha = 0.3$, $s = 0.8$), which results in a total of

**Figure 4 Ten new reactions found with barriers below 30 kcal/mol.** The stated barriers ($\Delta E^\dagger$) and reaction energies ($\Delta E$) are computed at the $\omega$B97X-D/def2-TZVP level of theory.

586 elementary reactions (average of 29.3 per reactant, Fig. S3). Among these reactions are 16 of the 20 target reactions but R129, R2191, R10077 and R7854 (Table S4) are not found. The lower success rate compared to unimolecular reactions (16/20 *vs* 23/26 for this hyperparameter set) is likely due to the increased number of reactions per reactants (29.3 *vs* 17.4 reactions per reactant on average). For a reactant system with only a single viable path (low-barrier single-step product) we expect a high probability that at least one of the 100 meta-MD simulations will go to that product. However, for a reactant

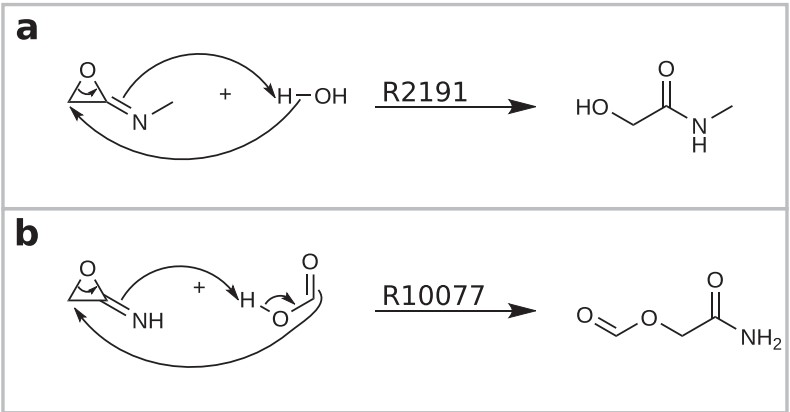

**Figure 5 (A) The reaction not found by any meta-MD hyperparameter sets (R2191).** The reaction energy is $\Delta E = -46$ kcal/mol and the barrier is $\Delta E^\dagger = 19$ kcal/mol calculated with $\omega$B97X-D3/def2-TZVP (*Grambow, Pattanaik & Green, 2020*). (B) Target reaction R10077 which is found with meta-MD using the secondary hyperparameter set, but where the GFN2-xTB barrier estimate from RMSD-PP is too high (60 kcal/mol). The reaction energy is $\Delta E = -39$ kcal/mol and the barrier is $\Delta E^\dagger = 10$ kcal/mol calculated with $\omega$B97X-D3/def2-TZVP (*Grambow, Pattanaik & Green, 2020*).

system with, say, 29 low-barrier single-step reactions there is a good chance that at least one of the 29 reactions will not be found by meta-MD in any of the 100 simulations. Assuming all 29 reactions are found with equal likelihood there is only a 40% chance that 100 meta-MD runs will find all 29 reactions, compared to a 96% chance for 17 low-barrier reactions. To account for this, we try to "encourage" the reactants to go to previously undiscovered products, by including products found by other MD simulations when computing the biasing potential. After filtering the 586 reactions found in the first set of runs with RMSD-PP estimated barriers, 318 reactions are left for DFT refinement. For the second set of runs (initiated with products from the first set of runs), after filtering based on both RMSD-PP barrier estimates and duplicates of reactions from the first run, 174 reactions are left for DFT refinement. We find one additional target reaction in this way: R129. Two (R7854 and R10077) of the remaining three target reactions can be found with meta-MD by changing the hyperparameter set to our secondary choice ($k_{push} = 0.03$, $\alpha = 0.7$ and $s = 0.6$).

With the three different kind of runs tested here we find 19 of the 20 target reactions shown in Table S4. The one reaction not found by meta-MD (R2191) is shown in Fig. 5A. Though R10077 (Fig. 5B) was found by meta-MD with our secondary hyperparameter set, it was predicted to have a too high barrier ($\approx 60$ kcal/mol) using the GFN2-xTB RMSD-PP estimate. We note that these two target reactions (Fig. 5) both involve oxiranimines and have similar mechanisms. Unlike the reactions causing problems in the search from single molecule reactants above, these two reactions have low reaction energies at the GFN2-xTB level of theory ($-37$ kcal/mol for R2191 and $-29$ kcal/mol for R10077).

The DFT refinement step localizes the 18 target reactions remaining at this point as well as 34 new reactions with barriers below 30 kcal/mol. Twenty-five of the new reactions

**Figure 6** The lowest-barrier reaction per reactant for the 12 reactants, where new reactions below 30 kcal/mol were found.

are located from the reactions found using our default hyperparameter set while the remaining nine reactions are found by penalising products found by the first meta-MD runs. The 34 new reactions are spread across 12 of the 20 reactants. Figure 6 shows the lowest-barrier reaction for each of these 12 reactants and the remaining 22 new low-barrier reactions can be found in Fig. S2. The new reactions represent a range of reaction energies

**Figure 7 Step in the synthesis of Berkeleyone A (*Elkin et al., 2017*).** This reaction was previously studied with imposed activation by *Lavigne et al. (2020)*.

between −47 and 20 kcal/mol and barriers ranging from 0.5 to 29 kcal/mol. We find both reactions where the two reactant molecules react with each other and unimolecular reactions where one of the reactant molecules goes through an isomerization reaction.

## Application to organic chemistry

The reactions tested thus far involve relatively small molecules, often with functional groups not usually seen in organic chemistry. We thus test our methodology on two reactions, one unimolecular and one bimolecular, that are more representative of those encountered in synthetic organic chemistry. Our goal here is simply to check whether the correct products can be found with the current hyperparameters rather than an exhaustive computational study of the reaction mechanisms.

### A unimolecular reaction

Inspired by *Lavigne et al. (2020)* we study an important step of the synthesis of Berkeleyone A reported by *Elkin et al. (2017)* (Fig. 7). We choose the same protonated epoxide reactant structure as Lavigne et al. for the starting point of our analysis (Fig. 7). In practice several protonation sites must be investigated but this process can easily be automated. Figure 8 highlights some pathways found using meta-MD for product generation and RMSD-PP for barrier estimates at the GFN2-xTB level. As our goal with this example is to check the ability of meta-MD + RMSD-PP to give insight into more complicated multi-step reactions compared to the elementary reactions studied until now we skip the DFT validation step and report the GFN2-xTB energies and barrier estimates. Thus, the energetics presented in Fig. 8 are not expected to be quantitatively accurate.

Doing meta-MD + RMSD-PP starting from the reactant structure (**R**) produces reactions involving ring-opening of the protonated epoxide to produce both secondary and tertiary carbocations, proton transfer from the epoxide to the carbonyl oxygen of the ester group, as well as tautomerization reactions. Instead of restarting the procedure from every one of the produced intermediates, we choose to follow the path involving the tertiary carbocation **I1** further. Continuing this process we create reaction profiles as presented in Fig. 8. We locate a possible mechanism for the path to the experimentally observed product (**P**) through the five steps connected in black. We also show a pathway to one of the other (less stable) stereoisomers of the product (**Pa**, blue) as well as the path to the most stable product encountered (**Pb**, purple). We note that both the intermediate **I2** and **Pb** is also found in the study conducted by Lavigne et al. We also find paths to macrocycle structures (from **I1**) which is also found by them. The most stable intermediate

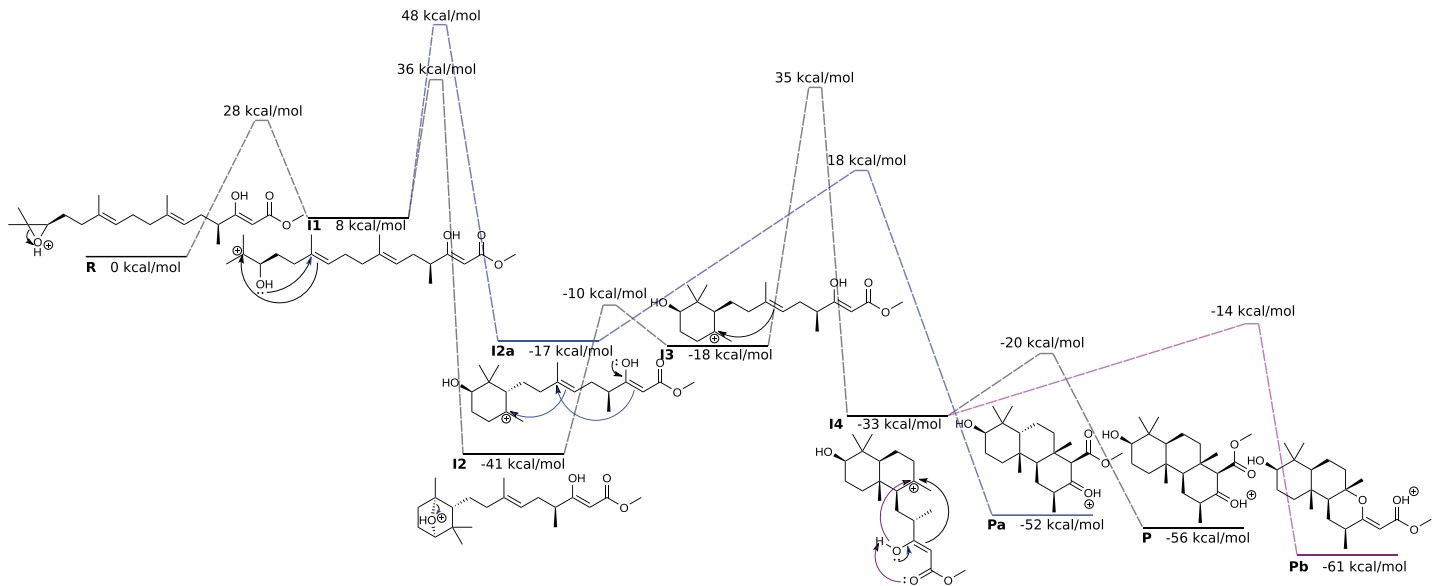

**Figure 8** **Some highlighted reaction paths found at GFN2-xTB level of theory.** Energies are relative to thereactant (**R**).

(**I2**), a bicyclic ether, is a known byproduct when doing this type of epoxide-initiated cyclizations (*Aggarwal, Bethel & Giles, 1999a*; *Aggarwal, Bethel & Giles, 1999b*).

The competition between carbon cyclisation (**P**) and oxygen cyclisation (**Pb**) is also a known problem for these reactions (*Aggarwal, Bethel & Giles, 1999a*; *Aggarwal, Bethel & Giles, 1999b*).

### A bimolecular reaction example

Next we study the acid catalyzed synthesis of a benzimidazole derivative starting from ortho-phenylene-diamine and benzoic acid (Phillips method) (*Phillips, 1928*).

The reaction is acid catalyzed so at each minimum along the path we try the different possible protonated structures and optimize with GFN2-xTB. Structures with energies <30 kcal/mol relative to the lowest energy protonation structure are considered relevant for the analysis. The result of the search is summarized in Fig. 9. For the reactant system, two protonation structures are possible: protonation at the nitrogen or at the carbonyl oxygen (which is 17 kcal/mol higher than at nitrogen). We follow the path from the carbonyl protonated structure and find step 1 by meta-MD (Fig. 9, step 1) where the lone pair of one of the nitrogen atoms is used to attack the carbonyl carbon. The next step is elimination of water which can be found in two ways: either by manually transferring the proton to one of the hydroxyl groups, resulting in water elimination upon energy minimization, or by adding a water molecule that can aid in the proton-transfer during the meta-MD simulation (Fig. 9, step 2). Step 3 is a ring closure initiated by attack of the other nitrogen atom. The second water elimination can again be found by manually moving the proton from the ammonium to the hydroxyl group (step 4) and subsequent geometry optimization which eliminates the water molecule (step 5) creating the product 2-phenyl benzamidazole. Contrary to the first water elimination, this second water elimination

**Figure 9 Summary of the reaction path found using meta-MD to follow the acid-catalyzed synthesis from ortho-phenylenediamine and benzoic acid to 2-phenyl benzamidazole.** Reaction energies and approximate barriers at GFN2-xTB level of theory.

was only found by manually moving the proton and not by the meta-MD runs with an additional water molecule (the backward reaction is the preferred path). The manual proton transfers can easily be automated.

## Timings

The CPU-time requirements for the semiempirical calculations are relatively modest compared to the DFT refinement calculations. A single meta-MD simulation requires on average 5.4 min on a single core of a Intel Xeon E5-2643 v3 (3.4 GHz) for the reactants of the low-barrier reaction dataset. Larger molecules such as the Berkeleyone A precursor (Fig. 7) require about 25 min, but the precise value depends greatly on how fast the reaction occurs; for the steps presented in Fig. 8 the average run time for a meta-MD simulation was in the range 17–35 min. A single semiempirical barrier estimate typically takes about 14 seconds on the same type of core (13 min for the Berkeleyone A precursor) and we usually run five of these in parallel with different settings. For comparison, a typical ωB97X-D/def2-TZVP TS search takes about 6.5 h on 2 cores.

## CONCLUSIONS

We test our meta-MD based approach for finding low-barrier (<30 kcal/mol) reactions for uni- and bimolecular reactions extracted from the barrier dataset developed by *Grambow, Pattanaik & Green (2020)*. Based on this dataset it should be possible to locate 26 low-barrier unimolecular reactions at the ωB97X-D/def2-TZVP level of theory starting from 163 reactants. Our method uses Grimme's meta-MD approach, with carefully chosen hyperparameters, to identify possible products, which are subsequently screened using semiempirical reaction energies and barrier heights before being refined with DFT (Fig. 1). The meta-MD simulations identify 25 of the 26 products found by *Grambow, Pattanaik &*

*Green (2020)*, while the subsequent semiempirical screening eliminates an additional four reactions due to an overestimation of the reaction energies or estimated barrier heights relative to DFT, suggesting that DFT may thus be needed in the screening process. In addition, our approach identifies an additional 36 reactions not found by *Grambow, Pattanaik & Green (2020)*, 10 of which have barriers <30 kcal/mol. All low-barrier reactions involve new reactants not represented in the low-barrier dataset, which indicates that the reactions in that dataset are likely the ones with the lowest possible barriers for each reactant.

*Grambow, Pattanaik & Green (2020)* only searched for elementary reactions starting from single reactant molecules, but they found a lot of products with two fragments. From these back-reactions, we extract a set of 20 target low-barrier reactions where two molecules react to create a single molecule (Table S4). While these reactions are not necessarily the ones with the lowest barrier for a given pair of reactant molecules, our method should be able to identify them along with any reactions with lower barriers. The meta-MD simulations identify 19 of the 20 products found by *Grambow, Pattanaik & Green (2020)*, while the subsequent semiempirical screening eliminates an additional reaction due to an overestimation of the barrier height relative to DFT. In addition, we find 34 new low-barrier reactions. We found that it is necessary to "encourage" the reactants to go to previously undiscovered products, by including products found by other MD simulations when computing the biasing potential as well as decreasing the size of the molecular cavity in which the MD occurs, until a reaction is observed.

The reactions in the *Grambow, Pattanaik & Green (2020)* data set involve relatively small molecules, often with functional groups not usually seen in organic chemistry. We thus test our methodology on two reactions, one unimolecular and one bimolecular, that are more representative of those encountered in synthetic organic chemistry, with the goal to simply check whether the correct products can be found with the current hyperparameters. The unimolecular reaction is a multi-step triple ring-closure (Fig. 7) an important step of the synthesis of Berkeleyone A reported by *Elkin et al. (2017)*. We locate a possible mechanism for the path to the observed product through the five steps, together with other known biproducts. The bimolecular reaction is the acid catalyzed syntheses of benzimidazole derivatives starting from ortho-phenylenediamine and benzoic acid (Phillips method) (*Phillips, 1928*), where we find all five steps in the generally accepted reaction mechanism. The meta-MD hyperparameters used in this study thus appear to be generally applicable to finding low-barrier reactions.

### Funding

This work was supported by VILLUM FONDEN (00022896). The funders had no role in study design, data collection and analysis, decision to publish, or preparation of the manuscript.

## Grant Disclosures

The following grant information was disclosed by the authors:
VILLUM FONDEN: 00022896.

## Competing Interests

Jan H. Jensen is an Academic Editor for PeerJ.

## Author Contributions

- Maria H. Rasmussen conceived and designed the experiments, performed the experiments, analyzed the data, performed the computation work, prepared figures and/or tables, authored or reviewed drafts of the paper, and approved the final draft.
- Jan H. Jensen conceived and designed the experiments, analyzed the data, authored or reviewed drafts of the paper, and approved the final draft.

## Data Availability

The code is available at GitHub: https://github.com/jensengroup/AutomatedReactionsMetaMD.

The data is available at Electronic Research Data Archive: https://erda.ku.dk/archives/07b98425436766cb7611d06a3cec2164/published-archive.html.

## Supplemental Information

Supplemental information for this article can be found online at http://dx.doi.org/10.7717/peerj-pchem.22#supplemental-information.

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
