# Peer review of "Fast and automated identification of reactions with low barriers using meta-MD simulations"

_PeerJ Physical Chemistry, doi:10.7717/peerj-pchem.22_

## Round 0.1 · original submission · Minor Revisions

Please find the supportive and very thorough reviews attached. These should support you refining the manuscript further. I am happy to follow the referees' overall assessment of minor revisions.

Reviewer 1 ·

Basic reporting

no comment

Experimental design

no comment

Validity of the findings

no comment

Additional comments

This is an excellent paper, methods described clearly, results should be of utility to the community.

Reviewer 2 ·

Basic reporting

The article is clear and well-referenced, with a suitable level of recognition of alternative strategies for reaction path finding. The article is well structured and logically laid out, and offers a self-contained narrative linking the aims, methods, results, analysis, and conclusions. The figures are clear and of generally high quality. Raw data is available via a link at the end of the manuscript.

Minor comments on the basic reporting of the manuscript are listed in the attached PDF.

Experimental design

The investigation carried out in this work is rigorous and to a high technical standard. The aims of the investigation are clearly set out, and the approach employed is mostly well-described for those who would choose to replicate their findings.

Minor comments on the experimental design are listed in the attached PDF.

Validity of the findings

All data used or generated in this work are provided, as are the scripts used to generate these data. The conclusions answer the questions proposed in the introduction, and are justified by the methods employed and the results presented. These conclusions are clear and are limited to the content of the work presented.

Minor comments on the validity of the findings are listed in the attached PDF.

Additional comments

The manuscript is well-written and well-presented. The scripts and data used and generated in this manuscript are openly available, however would benefit from more extensive documentation to facilitate their widespread use. The minor comments listed above are intended only to improve an already thorough piece of work. I do not believe any new primary data need to be generated, and many of my comments relate only to justification of the chosen methods and parameters rather than any fundamental issues with the work. I am therefore suggesting that only minor corrections are required before publication.

Annotated reviews are not available for download in order to protect the identity of reviewers who chose to remain anonymous.

·

Basic reporting

The manuscript is well written and enjoyable to follow. However, some aspects could be improved, making the article accessible to a broader scientific audience:

1) Figure 1 would benefit from including the explanation of abbreviations, and the meaning of colors used for the products (not all cultural backgrounds see red as unfavorable and green as positive). Also, I would suggest not using a combination of red and green concerning the colorblind readers.

2) Since the estimates with RMSD-PP method are critical in the screening stage of the workflow, it would be helpful to have a short, 1-2 sentences, explanation of its principle with its first mention.

3) On lines 142-143, adding info on the fact that the estimates come from RMSD-PP would make the distinctions between the estimates and the actual semi-empirical barriers easier.

4) Figure 9 would report the reaction path in a more comprehensive way when supplemented by data on reaction energies as well as barriers.

5) Figure S2 is missing the caption.

Experimental design

The dataset used is rather extensive, given the methodology used, and the extension towards also evaluating bimolecular reactions with meta-MD is valuable. Although, the description of the experimental procedures could be enhanced by providing more details on the following two issues:

1) on line 63, the authors provide an additional set of hyper-parameters used for meat-MD, which were found as in this study. However, there is no data on how were these parameters found, what other sets were used etc.

2) on line 77, the algorithm checks for changes in atomic connectivity every 5 ps. How is this performed? Do such checks rely on semi-empirical energy minimization as described on line 74?

Validity of the findings

I wonder if the authors have tried more meta-MD runs, in particular, for bimolecular reactions where the odds of enumerating all expected low-barrier reactions are relatively low. Given the computational costs, it should be feasible to markedly increase the number of runs at least for some of the problematic cases.

Reviewer 4 ·

Basic reporting

Overall, the manuscript is well-written. However, some phrases are in the abstract and conclusion 1:1 copies: “In addition, we find 34 new […] until a reaction is observed.” Some variation would be better here.

Mostly, the literature references also appear to be appropriate. Yet, in the beginning, a number of references is given in the context of “computational methods for exploring reaction space in an automated manner”. I suggest to add the paper from the Martínez group (https://doi.org/10.1038/nchem.2099) that also falls into this category. I think that it is worth noting that their “piston-like” pushing of molecules is a harsher predecessor to the here suggested cavity reduction.

Minor issues:
line 313 and abstract: “at an” -> “an”
line 87 "is" -> "are"
line 178 "GFN-xTB" -> "GFN2-xTB"

Experimental design

The used methodology is described quite well in chapter 2.
Due to the additional biases applied, the comp. cost is increased roughly by a factor of 3 compared to a minima search (cf. line 215)?

line 99-101: About testing only 5 structures. Is there a fall-back option, if more than just these 5 structures are found within a certain energy window?

line 126: To make the comparison with the previous work of Grambow better possible, the authors might consider to compute wB97X-D3 single-point calculations (for example, using ORCA) on the wB97X-D geometries.

line 284: an average atom number for the investigated systems should be given.

Validity of the findings

The data has been provided in an external repository, which is very good.
I personally would favor an (additional) upload together with the paper SI to have all the data available in the same place.

line 280: It is stated that the manual proton transfers could easily be automated. But the authors should comment on this - would an automatic procedure (exploring all paths) not grow dramatically in comp. cost?

Overall, the manuscript “Fast and automated identification of reactions with low barriers using meta-MD simulations” by Rasmussen and Jensen presents a computational approach to automatically identify low-energy barriers by means of a previously established and herein adjusted metadynamics approach. The adjustments include mainly adjusting the parameters of the RMSD potential, but all cross-usage of different products in the metadynamics simulation.
The results in this manuscript are of large interest to the computational chemistry community, as automated search for transition states is a, yet, unsolved problem. The presented approach will thus be of significance in automated determinations of reaction barriers that are used microkinetic modeling.

Additional comments

About line 126: A warning: wB97X, wB97X-D, wB97X-D3, and wB97X-V all have different XC functional parameters! So, wB97X-D3 is not simply the D3-corrected version of wB97X.

---

## Round 0.2 · accepted · Accept

Thank you for the revisions in alignment to the detailed reviews of your manuscript, which I find fully adequate and thus accept the manuscript for publication.